# New Stable Non-Vector Control Structure for Induction Motor Drive

**Pavol Fedor, Daniela Perdukova \***, **Peter Bober** and **Marek Fedor**

Department of Electrical Engineering and Mechatronics, Technical University of Kosice, Letna 9, 04200 Kosice, Slovakia; pavol.fedor@tuke.sk (P.F.); peter.bober@tuke.sk (P.B.); marek.fedor.4@student.tuke.sk (M.F.)
\* Correspondence: daniela.perdukova@tuke.sk; Tel.: +421-55-602-2254

**Abstract:** The article focuses on a design and experimental verification of continuous nonlinear systems control based on a new control structure based on a linear reference model. An application of Lyapunov's second method ensures its asymptotic stability conditions. The basic idea in the development of the control structure consists of utilizing additional information from a newly introduced state variable. The structure is applied for angular speed control of an induction motor (IM) drive representing a higher-order nonlinear system. The developed control algorithm helps to achieve the zero steady-state control deviation of the IM drive angular speed. Simulations and experiments performed in various operating states of the IM drive confirm the advantages of the new control structure. Except for set dynamics, the method ensures that the system is stable, invariant to disturbances, and is robust against variations of the parameters. When comparing the obtained control structure of the IM control with the classical vector control, the proposed control structure is simpler. In addition, the proposed control structure is linear, robust against variation in important parameters and invariant against external disturbances. The main advantage over conventional control techniques consists of the fact that the controller design does not require any exact knowledge of the system parameters and, moreover, it does not suffer from system stability problems. The method will find a wide applicability not only in the field of AC controlled drives with IM but also generally in control of industry applications.

**Keywords:** control system synthesis; induction motor; motion control; nonlinear control systems; variable speed drives; Lyapunov's second method

## 1. Introduction

Induction motors (IMs) are robust and reliable, and due to their low cost and maintenance, they find a wide utilization in industrial applications. A problem consists of their control requiring more complex control circuitry due to variable frequency, complex dynamics, and parameter variations [1–3]. The IM itself presents a typical example of a nonlinear and considerably oscillating system with incorporated positive feedback. The accuracy of its speed control is significantly influenced by unknown external disturbances and variable motor parameters.

Several control methods of the IM are known. The simplest one is the scalar speed control method. It has a simple control structure [4,5] due to which is suitable for simpler industrial applications. A better drive performance of the scalar control method requires using on account of a more expensive and less reliable solution.

A precise drive performance is obtained by field oriented control (FOC) employing classical cascade PI controllers [6]. Various modifications of the FOC usually requiring transformation of the IM variables into a rotating reference frame have been developed which enable to control the IM in a similar manner like a separately excited DC motor [7–9]. A drawback of such a solution consists of increased complexity of the control scheme

and in necessity of using powerful computational means (digital signal processors) for its implementation.

From the point of view of the modeling and control the IM principally presents a nonlinear dynamic system with uncertain parameters. Many design techniques for nonlinear control have already been applied for the control of IM drives providing better performance than the FOC [10,11].

Various sensorless methods to measure the rotor position of electrical drives with IMs [12–14] have been proposed in order to decrease the hardware complexity and cost and simultaneously to increase higher mechanical robustness and by this a reliability of the drive performance.

From the field of applications of nonlinear control methods to the speed and position control of the IMs, a sliding mode control should be mentioned [15–17]. Soft computing methods like fuzzy logic, artificial neural networks, evolutionary algorithms and their combinations have also been applied for the IM drive's control [18–23]. However, again, their drawbacks consists of powerful real-time calculation processes and, moreover, they can incline to the stability problems of the system (as is often the Z\case of fuzzy control theory methods).

An overview of several methods of the IM control at inaccurate determined parameters (the rotor resistance, mains failure and load torque) is given in [24]. It follows that the quality of the IM control depends in principle on the accuracy of the IM model used for all methods. However, the exact model of the drive is not available as at a large volume of production, the exact motor parameters can vary significantly.

Several methods have been proposed to solve the problem of IM control with an inaccurate model. A disadvantage of an adaptive version of the sliding mode [25,26] is that it guarantees the robustness only within a range of the uncertainties, and it still suffers from a chattering problem. The predictive control [27–29] can be complemented by a parameter observer to estimate the uncertain model parameters, but the stability for such schemes is usually not guaranteed. A backstepping control method [30], that has appeared recently, allows the design of the control law and the estimation of the motor parameters. However, the proposed method is suitable for a limited set of adaptive parameters only.

Summarizing the review, it is clear that the high quality of the IM control should take into account the following criteria: a nonlinear and oscillating character of the IM dynamics, variations of motor parameters, the influence of external disturbances, and a simple implementation of the control algorithm.

In this article, a new robust control structure with a reference model is designed to control the angular speed of an IM drive where the system stability is derived on basis of the Lyapunov's second method [31]. Its main advantage consists of the fact that that the design of the control structure does not require any accurate knowledge of values of the IM parameters. The resulting structure yields optimal dynamic properties in terms of the minimum control deviation and minimum input energy [32] criteria, which are normally used for evaluation of the control efficiency.

The proposed method consists of an extension of the control algorithm by aadditional information. This is easily obtained from the system output variable which ensures that the steady-state of the output variable is zero. If the control algorithm for the extended system is designed in such a way that the system would be asymptotically stable with the dynamics prescribed by the reference model, it will reach the goal of control both in the steady and in the transition states.

The properties of the proposed control structure have been verified by simulations and experimental measurements on the IM laboratory model. The proposed control structure is considerably simpler than an FOC structure. It is stable, linear, robust, and it has identical dynamical properties without any necessity for knowledge of an exact mathematical model of the IM. These features will increase the implementation potential of this strategy in industrial applications.

The paper is organized as follows: after the ntroduction, the design of the linear model reference control structure is presented in Section 2. Section 3 describes the mathematical model of IM, its parameters and properties, which in Section 4 are further used to design a control for the IM drive's angular speed control. The proposed control method is verified by simulation in various IM operating states and by experimental measurements on a laboratory model in Sections 5 and 7. Section 6 describes a comparison of the proposed control structure with the vector control structure (FOC). Finally, basic characteristics of the novel control structure are presented in Sections 8 and 9.

## 2. Design of the Linear Model Reference Control Structure

The desired system dynamics of a controlled system are very often described by a reference model. When this is a linear system, it can be optimally designed using standard methods of the optimal control theory [33]. The state-space reference model for the controlled system with $n$ state variables and $p$ inputs is described by the state equation:

$$\frac{\mathrm{d}\mathbf{x}_M}{\mathrm{d}t} = \mathbf{A}_M \mathbf{x}_M + \mathbf{B}_M \mathbf{w} \tag{1}$$

where $\mathbf{x}_M$ ($n \times 1$) is the state vector of the reference model, $\mathbf{A}_M$ ($n \times n$) is the state matrix of the reference model, $\mathbf{B}_M$ ($n \times p$) is the input matrix of the model and $\mathbf{w}$ ($p \times 1$) is the vector of the desired values.

The controlled system is described in state space as a nonlinear continuous system with parametric and with additive disturbances (or deviations from the reference model) in the form:

$$\frac{\mathrm{d}\mathbf{x}}{\mathrm{d}t} = (\mathbf{A}_M + \Delta\mathbf{A})\mathbf{x} + (\mathbf{B}_M + \Delta\mathbf{B})\mathbf{u} + \mathbf{v} = \mathbf{A}_M\mathbf{x} + \mathbf{B}_M\mathbf{u} + (\Delta\mathbf{A}\mathbf{x} + \Delta\mathbf{B}\mathbf{u} + \mathbf{v}) \tag{2}$$

where $\mathbf{x}$ ($n \times 1$) is the state vector of the controlled structure, $\mathbf{u}$ ($p \times 1$) the vector of input variables, $\Delta\mathbf{A}$ ($n \times n$), $\Delta\mathbf{B}$ ($n \times p$)—the matrices of the parametric disturbances and $\mathbf{v}$ ($n \times 1$)—the vector of additive disturbances.

In this case, the goal of the electric drive control is twofold:

(1) To reach the zero state of the state vector $\mathbf{x}$.
(2) To reach the zero state of all deviation of state variables from the desired values.

For this reason, the deviations of the state vector components from the desired values are suitable to be chosen as the state variables of the controlled system. The system's stability is investigated with regard to these deviations.

Let us calculate the deviation between the reference model and controlled system:

$$\mathbf{e} = \mathbf{x}_M - \mathbf{x} \tag{3}$$

where $\mathbf{e}$ ($n \times 1$) is the vector of deviations between the state variables of the model $\mathbf{x}_M$ according to (1) and of the system $\mathbf{x}$, defined in Equation (2). By differentiating this vector of deviations one gets:

$$\frac{\mathrm{d}\mathbf{e}}{\mathrm{d}t} = \frac{\mathrm{d}\mathbf{x}_M}{\mathrm{d}t} - \frac{\mathrm{d}\mathbf{x}}{\mathrm{d}t} \tag{4}$$

After inserting Equations (1) and (2), the expanded system is:

$$\frac{\mathrm{d}\mathbf{e}}{\mathrm{d}t} = \mathbf{A}_M(\mathbf{x}_M - \mathbf{x}) + \mathbf{B}_M\mathbf{w} - \mathbf{B}_M\mathbf{u} - \Delta\mathbf{A}\mathbf{x} - \Delta\mathbf{B}\mathbf{u} - \mathbf{v} \tag{5}$$

$$\frac{\mathrm{d}\mathbf{e}}{\mathrm{d}t} = \mathbf{A}_M\mathbf{e} - \mathbf{B}_M\mathbf{u} + \mathbf{f} \tag{6}$$

where the vector $\mathbf{f}$ ($n \times 1$) presents a generalized disturbance vector comprising all parametrical and additive disturbances affecting the system with regard to its reference model:

$$\mathbf{f} = -\Delta\mathbf{A}\mathbf{x} - \Delta\mathbf{B}\mathbf{u} - \mathbf{v} + \mathbf{B}_M\mathbf{w} \tag{7}$$

The goal of the controller design is to find such mathematical formulation for determining the input vector $\mathbf{u}$ for which the zero solution of the system (6) is asymptotically stable, i.e., $\lim\limits_{t\to\infty}\mathbf{e} = 0$. In order to investigate the asymptotic stability of the system (6) according to the Lyapunov criterion, the positive definite Lyapunov function is chosen in the weighted quadratic form of the system states:

$$V = \mathbf{e}^T\mathbf{P}\mathbf{e} \tag{8}$$

The derivation of the Lyapunov function (8) after inserting (1), (2), (3), (6) and performing simple modifications is:

$$\frac{\mathrm{d}V}{\mathrm{d}t} = \mathbf{e}^T\left(\mathbf{A}_M^T\mathbf{P} + \mathbf{P}\mathbf{A}_M\right)\mathbf{e} + 2\left[\mathbf{f}^T\mathbf{z} - (\mathbf{B}_M\mathbf{u})^T\mathbf{z}\right] = \mathbf{e}^T\mathbf{Q}\mathbf{e} + 2\left[\mathbf{f}^T\mathbf{z} - (\mathbf{B}_M\mathbf{u})^T\mathbf{z}\right] \tag{9}$$

where the vector $\mathbf{z}$ ($n \times 1$) is the weighted state deviation vector:

$$\mathbf{z} = \mathbf{P}\mathbf{e} \tag{10}$$

In (8)–(10) the matrix $\mathbf{P}$ ($n \times n$) is a symmetric positively definite matrix which satisfies the Lyapunov matrix equation:

$$\mathbf{A}_M^T\mathbf{P} + \mathbf{P}\mathbf{A}_M = -\mathbf{Q} \tag{11}$$

where $\mathbf{Q}$ ($n \times n$) is also a symmetric positive definite matrix.

Choosing the reference model according to (1) one can avoid solving (11). If the state matrix of the reference model is in the controllability form, then based on the optimal control theory [32] it is possible to determine the elements of the matrix $\mathbf{P}$ analytically as follows:

$$\mathbf{Q} = -\alpha\mathbf{P} \tag{12}$$

where the parameter $\alpha$ allows to set an optimal dynamics of the controlled variable satisfying the criteria of the minimum control deviation and of the minimum input energy. The model dynamics are inversely proportional to the value of the parameter $\alpha$.

The system (6) is asymptotically stable if the derivation of Lyapunov function (9) is a negative definite function. Based on (10)–(12), the derivation of the Lyapunov function is:

$$\frac{\mathrm{d}V}{\mathrm{d}t} = -\alpha\mathbf{e}^T\mathbf{P}\mathbf{e} + 2\left[\mathbf{f}^T\mathbf{z} - (\mathbf{B}_M\mathbf{u})^T\mathbf{z}\right] \tag{13}$$

$$\frac{\mathrm{d}V}{\mathrm{d}t} = -\alpha\mathbf{e}^T\mathbf{z} + 2\left[\mathbf{f}^T\mathbf{z} - (\mathbf{B}_M\mathbf{u})^T\mathbf{z}\right] \tag{14}$$

Here, the expression $\mathbf{e}^T\mathbf{z} = \mathbf{e}^T\mathbf{P}\mathbf{e}$ (where $\mathbf{z} = \mathbf{P}\mathbf{e}$) is always positive and as a result, the term $-\alpha\mathbf{e}^T\mathbf{z}$ in (14) is negative. Then, the system (6) will be asymptotically stable, i.e., its derivation will be negative, if for the input $\mathbf{u}$ it holds that:

$$\mathbf{u} = \mathbf{K}\mathbf{z} \tag{15}$$

where $\mathbf{K}$ ($n \times n$) is an optional constant matrix of positive parameters. The matrix $\mathbf{B}_M$ is a constant matrix and its influence can be generally included in the values of the optional elements of the matrix $\mathbf{K}$, when modifying Equation (14) with respect to Equation (15).

Then, the second expression on the right side in (14) will be negative if for each component of the vector **f** the following inequality is met:

$$\left| \mathbf{k}_i^T \mathbf{z} \right| \geq |f_i| \quad \text{pre } i = 1 \dots n \tag{16}$$

where $\mathbf{k}_i^T$ is the $i$-th row of the matrix **K**. Let us note that for a single-input system instead of the matrix **K** the row vector $\mathbf{k}^T$ is used. The inequality (16) is ensured if the optional positive parameters in the matrix **K** will have sufficiently large values.

In order to achieve the zero control deviation, i.e., the difference between the output variables of the reference model and the controlled system ($y_M - y$) in steady-state, the first component of the vector **e** will be chosen as an integral of this difference:

$$x_{ext} = \int (y_M - y) \mathrm{d}t \tag{17}$$

By introducing this integral, the reference model (1) is extended by the new state variable $x_{ext}$. Now, the extended deviation vector (3) **e*** will be:

$$\mathbf{e}^* = \left[ \begin{array}{c} x_{ext} \\ \mathbf{e} \end{array} \right] \tag{18}$$

The reference model extension does not affect the stability of the designed control structure provided that the conditions in Equations (10)–(12) are valid also for the extended system.

Maximum values of the disturbance vector components |**f**| usually are physically limited. The limitation can be ensured by a relevant increase of the values of the optional parameters in the matrix **K**—the condition (16).

The block diagram of the designed controlled system with the extended reference model derived according to the above theory is shown in Figure 1.

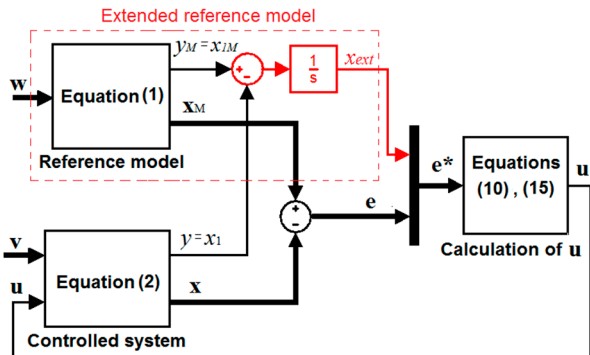

**Figure 1.** The structure of the designed controlled system.

The following basic features result from the control structure design:

- The control structure design and controller parameters do not depend on accurate values of the nonlinear system parameters and thus, they do not depend on an exact description and the values of the nonlinear controlled system parameters;
- The controlled system dynamics are prescribed by the reference model (1), where both the reference model and the controlled system are of the same order. The linear reference model can be designed using standard methods of linear control theory in order to set an optimal motion dynamics of the controlled system;
- The reference model dynamics can be set using a single parameter $\alpha$. By its introduction, the matrix **P** satisfies the Lyapunov matrix Equation (11) and according to [32] it does not have to be solved;

- The stability of the control is ensured by condition (16), in which the positive elements (gains) of the matrix **K** present the optional parameters at the controller design.
- The values of the disturbance vector **f** components in technical systems usually are physically limited. It means that at sufficiently large values of the matrix **K** elements, the control deviation of the output (controlled) variable always converges to zero in steady-state.

## 3. Mathematical Model of Induction Motor

The complete mathematical model of the IM presents a nonlinear dynamic system of the 5th order. In the following, it is assumed that the current control loops are involved in circuits of a standard frequency converter. Due to their high dynamics, their influence is neglected. Then, the current–flux model of the IM presents the 3rd order system. Its current–flux model [8] is described in the $\{x, y\}$ reference system by components of the stator currents and rotor fluxes:

$$\frac{d\psi_{2x}}{dt} = -\omega_g \psi_{2x} + L_m \omega_g i_{1x} + (\omega_1 - \omega)\psi_{2y} \tag{19}$$

$$\frac{d\psi_{2y}}{dt} = -\omega_g \psi_{2y} + L_m \omega_g i_{1y} - (\omega_1 - \omega)\psi_{2x} \tag{20}$$

$$T_{mech} = L_m \frac{p}{L_1}\left(\psi_{2x} i_{1y} - \psi_{2y} i_{1x}\right) \tag{21}$$

$$\frac{J}{p}\frac{d\omega_m}{dt} = T_{mech} - T_L \tag{22}$$

where the notation of the parameters and variables is as follows:

$i_{1x}, i_{1y}$    components of the stator current space vector $\mathbf{i}_1$;
$\omega_m$    mechanical angular speed of the rotor;
$\omega_1$    angular frequency of the stator voltage;
$\omega_2$    slip angular speed $\omega_2 = \omega_1 - \omega_m$;
$r_2$    rotor phase resistance;
$r_1$    stator phase resistance;
$\psi_{2x}, \psi_{2y}$    stator and rotor magnetic flux components;
$L_m$    main inductance;
$L_1, L_2$    leakage inductances;
$\Omega_g$    constant $\omega_g = R_2/L_2$;
$T_{mech}$    mechanical motor torque;
$T_D$    dynamic motor torque;
$p$    number of pole pairs;
$J$    moment of inertia;
$T_L$    load torque.

From Equations (19)–(22) it is obvious that the IM presents a strongly nonlinear higher-order controlled system of an oscillating character.

The values of the motor parameters used for simulation are specified in Table 1.

**Table 1.** Induction motor parameters for modeling and experimentation.

| | | |
|---|---|---|
| $P_N = 3$ kW | $U_{1N} = 220$ V | $I_{1N} = 6.9$ A |
| $J = 0.1$ kgm$^2$ | $r_1 = 1.8\ \Omega$ | $R_1 = 2/3\ r_1 = 1.2\ \Omega$ |
| $n_N = 1430$ rev./min | $r_2 = 1.85\ \Omega$ | $R_2 = 2/3\ r_2 = 1.23\ \Omega$ |
| $p = 2$ | $L_1 = L_2 = 0.2106$ H | $\omega_g = R_2/L_2 = 5.84$ s$^{-1}$ |
| $T_N = 20$ Nm | | |

The characteristics in Figure 2 show the motor model torque and angular speed responses at the step change on the motor inputs in time $t = 0$ s, when the motor is

supplied from a current converter in which the current loop time constant already is compensated. Let us note that according to (19)–(22) in this case, the values of the current vector components of the IM model used for simulation are $i_{1x} = 0$ A, $i_{1y} = 15$ A. At this supply, the angular speed reaches the value of $\omega_1 = 200$ rad/s.

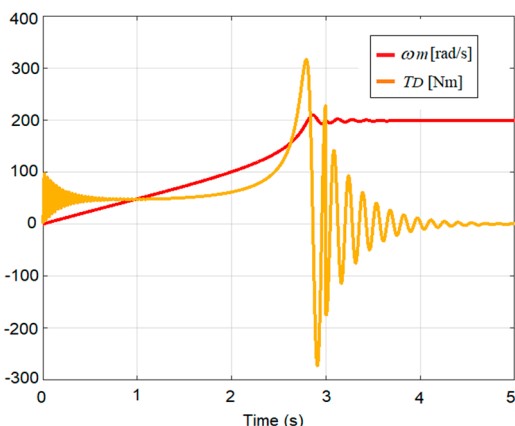

**Figure 2.** Time courses of the IM model torque and speed at motor starting when supplied by the current vector.

## 4. Design of the IM Drive Control

The objective of the IM drive control consists of control of its angular speed where the motor dynamics are prescribed by a linear reference model to be designed. For the control structure design according to Section 2, it is not necessary to know an exact model of the controlled system. The electromagnetic phenomena within the motor are considerably faster than the mechanical phenomena on the motor shaft and the electrical circuits comprise all nonlinearities. The electromagnetic phenomena dynamics are replaced by the 1st order proportional system and the mechanical phenomena are similarly replaced by the 1st order system. The simplified model of the IM presents a nonlinear dynamic system of the 2nd order as shown in Figure 3.

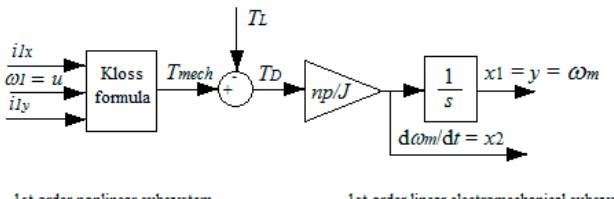

1st order nonlinear subsystem          1st order linear electromechanical subsystem

**Figure 3.** Simplified model of the IM presenting a 2nd order nonlinear system.

The static nonlinearity in generating the mechanical torque of the $T_{mech}$ motor is usually described by the Kloss formula with the 1st order dynamics and the electromechanical linear subsystem is described by the equation of the torque equilibrium on the drive shaft.

Let us select the state variables of the IM as shown in Figure 3: $x_1 = y = \omega_m$ (the rotor angular speed) and $x_2 = dx_1/dt = d\omega_m/dt$ (the rotor acceleration corresponding to its motor dynamic torque $T_D$).

As the IM is considered as a nonlinear 2nd order system, according to [32] its required dynamics are prescribed by the 2nd order linear reference model:

$$\begin{bmatrix} \dot{x}_{1M} \\ \dot{x}_{2M} \end{bmatrix} = \begin{bmatrix} 0 & 1 \\ -\frac{\alpha^2}{2} & -\alpha \end{bmatrix} \begin{bmatrix} x_{1M} \\ x_{2M} \end{bmatrix} + \begin{bmatrix} 0 \\ \frac{\alpha^2}{2} \end{bmatrix} w \qquad (23)$$

where the state variables $x_{1M}$ and $x_{2M}$ prescribe the dynamic behavior of the corresponding state variables of the controlled system, i.e., $x_{1M} = y_M = \omega_{mM}$ and $x_{2m} = dx_{1M}/dt = d\omega_{mM}/dt$.

The optimal dynamics of the controlled variable can be set by the optional positive parameter $\alpha$ in the reference model (23).

To ensure the zero control deviation of the angular speed of the controlled system and the reference model at steady-state, the reference model is extended by a new state variable $x_{ext}$.

To create a deviation between the state variable of the reference model and the system according to (17), the following relation is used:

$$x_{ext} = \int (x_{1M} - x_1)dt = \int (y_M - y)dt \tag{24}$$

Now the extended vector of the deviation will be in the form:

$$\mathbf{e}^* = \begin{bmatrix} x_{ext} \\ e_1 \\ e_2 \end{bmatrix} \tag{25}$$

This means the extended reference model of the IM drive presents the 3rd order system. According to the optimization theory [32], its state matrices are in the form:

$$\mathbf{A}_M = \begin{bmatrix} 0 & 1 & 0 \\ 0 & 0 & 1 \\ -\frac{\alpha^3}{2} & -\frac{3\alpha^2}{2} & -\frac{3\alpha}{2} \end{bmatrix}; \quad \mathbf{b}_M = \begin{bmatrix} 0 \\ 0 \\ \frac{\alpha^2}{2} \end{bmatrix} \tag{26}$$

where $\mathbf{b}_M$ is the input vector in the case of the one-input model.

The optimal matrices **P** and **Q**, that satisfy the Lyapunov matrix Equation (11), are:

$$\mathbf{P} = \begin{bmatrix} \frac{\alpha^5}{2} & \alpha^4 & \frac{\alpha^3}{2} \\ \alpha^4 & \frac{5\alpha^3}{2} & \frac{3\alpha^2}{2} \\ \frac{\alpha^3}{2} & \frac{3\alpha^2}{2} & \frac{3\alpha}{2} \end{bmatrix}; \quad \mathbf{Q} = -\alpha\mathbf{P} \tag{27}$$

The IM as a controlled system will follow the extended reference model with $\lim\limits_{t \to \infty} \mathbf{e}^* = \mathbf{0}$, i.e., the controlled system will be asymptotically stable, if the input $u$ is calculated using (15):

$$u = \begin{bmatrix} k_1 & k_2 & k_3 \end{bmatrix} \begin{bmatrix} z_1 \\ z_2 \\ z_3 \end{bmatrix} \tag{28}$$

According to (16), the elements for the vector $\mathbf{k}^T$ presenting gains must be positive and large enough to ensure the asymptotic stability of the controlled system. On the other hand, the value of the parameters in the vector $\mathbf{k}^T$ is limited by physical constraints in the controlled system. In the case of an electric drive, it is the electric motor current, the dynamics of real power converter, etc.

According to (10), the components of the vector $\mathbf{z}$ are:

$$z_1 = \mathbf{p}_{11}x_{ext} + \mathbf{p}_{12}e_1 + \mathbf{p}_{13}e_2 \tag{29}$$

$$z_2 = \mathbf{p}_{21}x_{ext} + \mathbf{p}_{22}e_1 + \mathbf{p}_{23}e_2 \tag{30}$$

$$z_3 = \mathbf{p}_{31}x_{ext} + \mathbf{p}_{32}e_1 + \mathbf{p}_{33}e_2 \tag{31}$$

The elements of the matrix **P** are known from evaluating Equation (27).

The resulting control scheme for controlling the angular speed of the IM drive in accordance with the derived control structure presented in Figure 1 is shown in Figure 4.

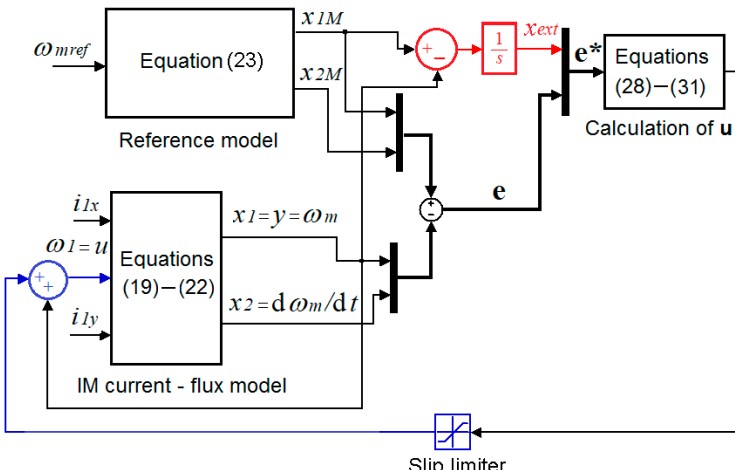

**Figure 4.** IM drive block diagram.

In order for the IM to operate in a stable part of its torque characteristic, the block diagram in Figure 4 has been completed by a conversion of the action variable to slip that is completed by its limitation (like it is case in a real motor). The slip limiter, however, does not change the validity of the structure shown in Figure 1. The limitation of the state variables of the drive system (the current and torque) can be realized inside the reference model without any affecting the control loop stability. This is due to the fact that the reference model is linear and the limitations do not influence the position of the linear system poles.

## 5. Simulation of the IM Drive in Basic Operation States

The verification of the control structure properties was carried out by simulation in MATLAB with the motor parameters specified in Table 1. The desired value of the mechanical angular speed of the IM was set in the reference model to $\omega_{mref}$ = 150 rad/s and the stator current vector components were $i_{1x}$ = 0 A, $i_{1y}$ = 20 A.

The proposed controller structure is based on Lyapunov's second method, which defines the range of its optional parameters, i.e., values of the elements of the matrix **P** and the positive optional vector of parameters $\mathbf{k}^T$ for which the controlled system is stable.

The elements of the matrix **P** are computed from the Lyapunov matrix Equation (11), where the positive definite matrix **Q** must be chosen. The values of elements of the matrix **Q** (and thus of the matrix **P**) influence the speed of decreasing the Lyapunov function (Equation (8)), which means a deceleration rate of the control deviation $e$. The system (6) will be stable for any arbitrarily selected elements of the matrix **Q** when fulfilling the condition of its positive definiteness. However, if we choose the reference model which ensures the optimal dynamic properties of the controlled system according to the criteria of minimal control deviation and minimum input energy, then it is possible to avoid solving the Lyapunov matrix Equation (11), because the elements of the matrix **P** can be determined analytically, based on Equation (12) from the matrix (27) (see [32]):

$$\mathbf{P} = \begin{bmatrix} 1562.5 & 625 & 62.5 \\ 625 & 312.5 & 37.5 \\ 62.5 & 37.5 & 7.5 \end{bmatrix} \tag{32}$$

The designed controlled structure also includes the controller gain, which is represented by optional elements of the vector $\mathbf{k}^T$ in Equation (28) for calculating the input $u$. These elements must be positive and their value must be chosen in such a way that during the operation the IM must not reach the current limit. To fulfil all requirements, the vector $\mathbf{k}^T$ in (28) has been set to the value $\mathbf{k}^T$ = [0.0031 0.0019 0.00038]. These values were obtained by a gradual increasing the values of individual elements of the vector $\mathbf{k}^T$ so that the

dynamics of decreasing the Lyapunov function would be the largest one at considering physical limits of the control system.

An advantage of the described settings of parameters against the classical approaches consists of the fact they can be determined from measurements of responses and form the knowledge of the system limits without knowledge of its precise parameters (e.g., of the rotor resistance, moment of inertia, etc.).

The dynamics of the reference model setting (using the parameter $\alpha$) must take into account the physical properties of the drive, which in our case are presented by the current (torque) overloading of the IM. The dynamic properties of the IM are described by unit step characteristics shown in Figure 2. The value of the parameter $\alpha$ is inversely proportional to the model dynamics. This means that based on the Shannon–Kotelnik theorem for stabilizing the angular speed of the IM within 1 s, the value of the optional parameter is chosen as $\alpha = 5$.

The operation cycle of the IM drive consists of three phases: starting the IM drive, running at a constant speed, and stopping (Figure 5a). An external (additive) disturbance of the IM drive—the load torque $T_L = T_N = 20$ Nm—was also introduced in time $t = 3$ s. This is valid for all simulation experiments.

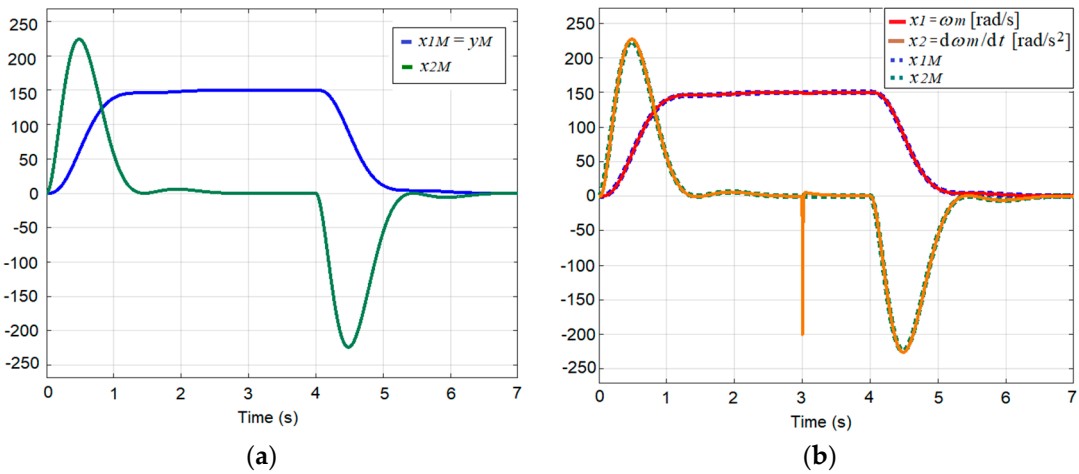

**Figure 5.** Time courses of the IM control during the operation cycle: (**a**) prescribed by the reference model, (**b**) achieved by the motor at its starting and loading.

The time course of the angular speed $\omega_m$ (the output variable) for the considered operation cycle is shown in Figure 5. It is obvious that the angular speed of the IM drive practically tracks the reference angular speed prescribed by the reference model (Figure 5a) during the entire operation cycle, even during the step disturbance $T_L$ as is shown in Figure 5b. This is also observed on changes of the variable $d\omega_m/dt \approx T_D$ (corresponding to the acceleration) in Figure 5b, where the considered disturbance torque $T_L$ is settled with high dynamics. This experiment verifies that the proposed control structure is invariant against the additive disturbances.

The robustness of the proposed control structure has been verified at the change of the two most important parameters of the controlled system significantly affecting its properties. In the case of IM, they are the rotor resistance $R_2$ and the moment of inertia $J$.

In the majority of the control systems for IM, the precise speed control depends on knowledge of the rotor resistance value that usually is identified by various types of observers. Figure 6 shows the time response when the rotor resistance $R_2$ was increased to twice its original value. The dynamics of the controlled system are almost identical to the dynamics of the reference model during the entire operation cycle (Figure 5b). The time courses of the variables in Figure 6 confirm the controlled system robustness for the considered variation of the parameter $R_2$.

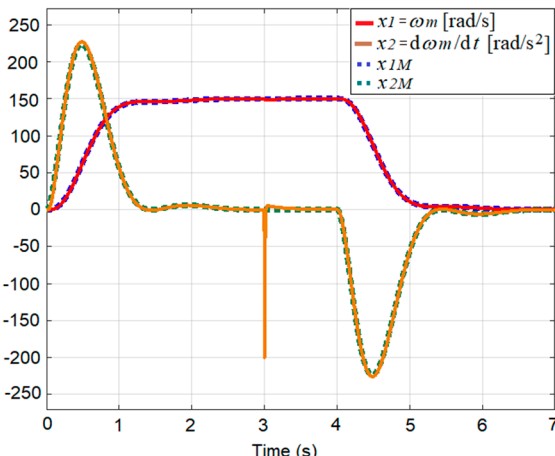

**Figure 6.** The effect of change of rotor resistance $R_2 = 2\,R_{2N}$ on control dynamics at motor starting and loading.

Another parameter that considerably influences the motor dynamics is the moment of inertia $J$ applied to the motor shaft. Its value can vary during the operation cycle, which is the case for many industrial applications of the IM drives (e.g., winding machines, robotic and transportation systems). As it follows up from the time courses in Figure 7a,b, the increase of the moment of additional inertia to a half/or double that of the motor's nominal moment of inertia does not influence the motor dynamics. This fact presents a very significant advantage in the control of nonlinear systems with an oscillating character. The effect of the load torque $T_L$ is again compensated with high dynamics also at a significant change of the motor inertia, which confirms the high robustness against the parameter variations in the proposed control structure.

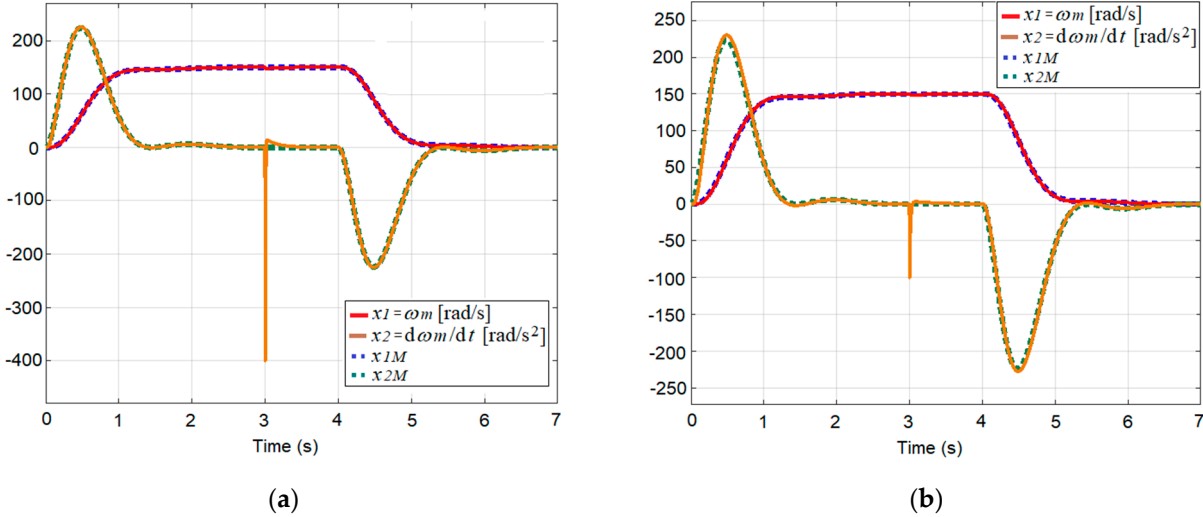

(**a**)  (**b**)

**Figure 7.** The effect of change of additional moment of inertia to the value: (**a**) $J = 0.5\,J_N$; (**b**) $J = 2\,J_N$ on control dynamics at motor starting and loading.

The simulations have confirmed that the proposed controller ensures a high quality angular speed control of the IM drive during the entire operation cycle according to the prescribed dynamics. Moreover, the controller satisfies all basic control objectives: the drive is invariant against the disturbances (the load torque) and robust against changes of the motor parameters (the rotor resistance and additional moment of inertia connected to the rotor shaft).

## 6. Comparison of the Proposed Control Structure with the Vector Control Structure

The vector drive control of the IM presents one of the most common control methods in technical practice today. Its specific dynamic properties depend on the used modification and are described by a large amount of the literature, e.g., [6,10]. At first, the proposed control properties will be compared with those of the vector control on the basis of a comparison of their structures. Secondly, the performance comparison of these two controlled structures will be given.

The basic structures of both control methods are shown in Figures 8–10. In both structures, the upper control level controls the desired mechanical speed $\omega_m$ of the drive, and eventually it sets the excitation magnetic flux magnitude of the motor through its current components in the rectangular rotating reference frame $\{x, y\}$ rotating at the angular speed $\omega_1$.

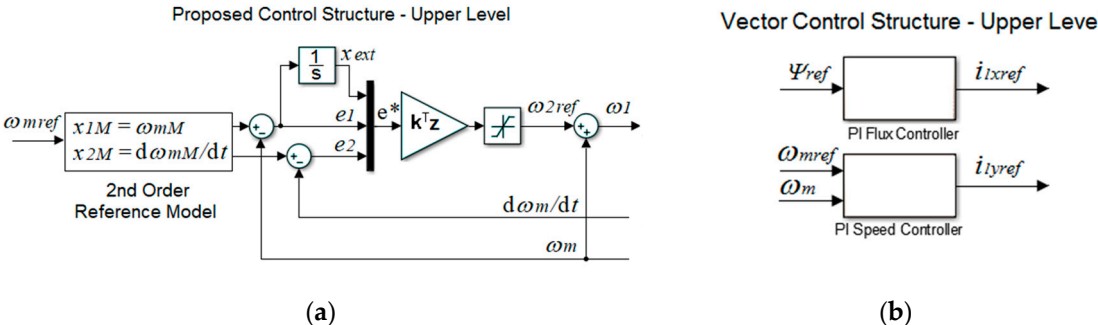

(**a**)                                                                                    (**b**)

**Figure 8.** The upper control level: (**a**) of the designed control structure; (**b**) of the vector control structure.

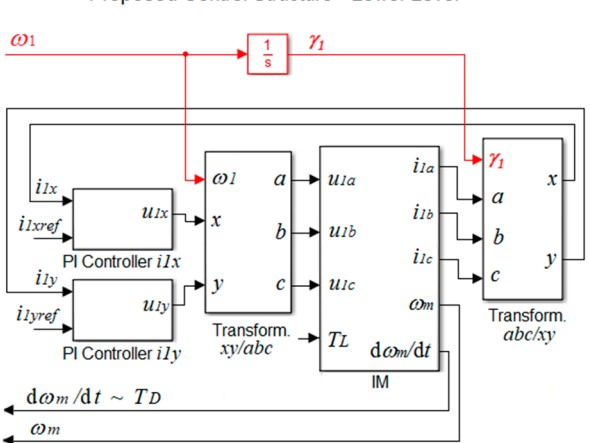

**Figure 9.** The lower control level of the proposed control structure, where $\gamma_1$ is the instantaneous position of the stator voltage vector and $u_{1x}$, $u_{1y}$ are the components of the stator voltage space vector $\mathbf{u}_x$.

At the upper control level (Figure 8a,b), there are two basic differences:

1. The first difference relates to the control structures: both structures are linear. They differ in their internal interconnection and by the control structure order;
2. The second difference consists of the type of the output action variable. In the vector control, they are variable setpoints of the stator current components $i_{1x}$, $i_{1y}$. In the proposed control structure the angular speed of the stator current vector presents the action variable and, in principle, the components $i_{1x}$, $i_{1y}$ are kept constant.

The lower control level (Figure 9) contains the controllers of the stator current components $i_{1x}$, $i_{1y}$ and the transformations between the stator system $\{a, b, c\}$ and the reference

frame $\{x, y\}$. At the lower control level, the two structures being compared are practically identical.

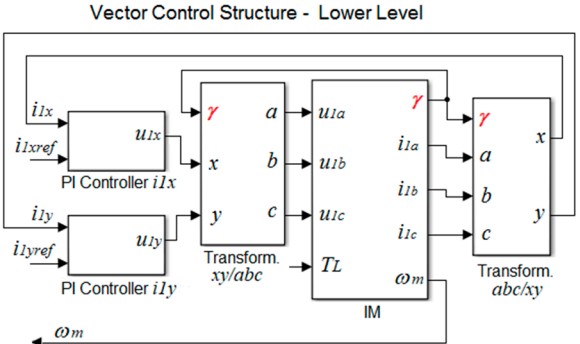

**Figure 10.** The lower level of the vector control structure, where $\gamma$ is the actual position of the rotor flux vector.

A significant difference consists of the stator currents transformations into the reference frame $\{x, y\}$. While in the vector control, this system most often tied to the position of the rotor flux vector $\gamma$, in the proposed control structure it is firmly tied to the position of the stator current vector $\mathbf{i}_1$ (which in the fact presents an integral from the angular speed $\omega_1$). In the case of the vector control (Figure 10), the problems arise when using the Park transformation. It requires the measurability of the rotor flux position, the dependence of this position estimation on the unknown and variable value of the rotor resistance, problems at low mechanical angular speeds, etc. These uncertainties can significantly affect the quality of the control.

In the proposed control structure, the value of the stator angular velocity $\omega_1$ inputs from the higher control level. Its value is always precisely known which guarantees the accuracy of this transformation and its independence from the current states or other parameters of the IM.

A basic comparison of the drive's behavior with closed control loops for exactly known IM parameters is shown in Figure 11. The first two figures from the left compare the response of the closed control loop to a step of the controlled system setpoint and to a disturbance. Here, the motor is starting, running to the angular speed $\omega_m = 150$ rad/s, then in time $t = 3$ s it is loaded by the nominal load torque $T_L = T_N = 20$ Nm and finally, it is stopping.

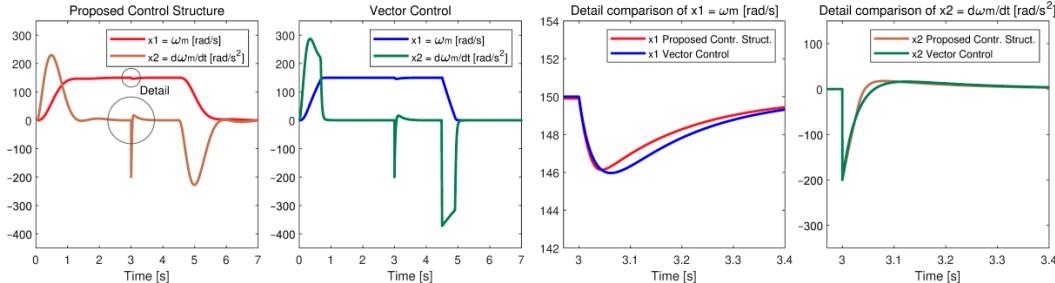

**Figure 11.** Comparison of dynamic properties of the controlled drive with the induction motor: the proposed and the vector control structures.

The next two figures show a detail of the course of the first and second state variables of the controlled system, i.e., the angular velocity $\omega_m$ and the dynamic torque $\omega_m/dt$ when the IM drive is loaded by the nominal load torque in time $t = 3$ s. In the case of vector control, the dynamic torque of the motor at its starting and stopping is higher. Its course has a non-linear shape, while in the proposed structure, the IM drive behaves as a linear 2nd order system. The time response to the load torque in the dynamic state is practically



the same, which is shown in Figure 11 in details of individual courses of the motor state variables.

Figure 12 compares the effect of an unknown change of the rotor resistance $R_2$ on the dynamics of closed control loops for two cases: for the designed control structure and for the vector control. While the dynamics of the motor with the proposed control structure remain almost unchanged, in the case of vector control, the change of the parameter $R_2$ during starting and stopping leads to a visible destabilization of the motor dynamic torque, both during the starting and especially during its loading. For the reduced value of the rotor resistance ($R_2 = 0.5 \, R_{2N}$), the dynamics of the vector control are worse than for the proposed control.

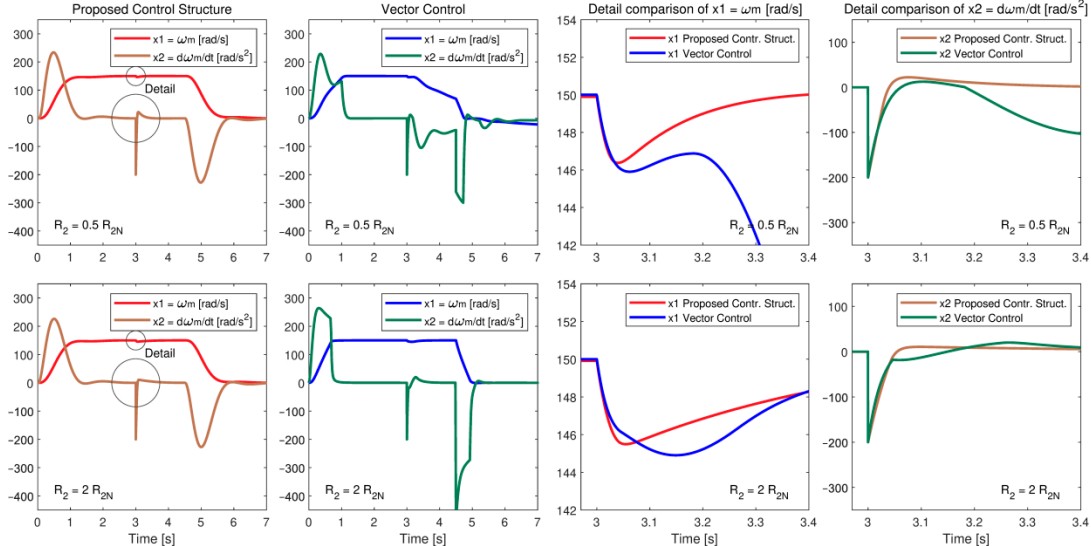

**Figure 12.** Comparison of the proposed and the vector control structures for change of the rotor resistance to the values $R_2 = 0.5 \, R_{2N}$ and $R_2 = 2 \, R_{2N}$.

The effect of an unknown change of the moment of inertia on the motor shaft on the dynamics of the closed control loops is compared in graphs in Figure 13. While the dynamics of the proposed control structure during the motor starting and stopping behaves like a linear 2nd order system, the vector control significantly changes its dynamic properties (mainly the settling time). From the point of view of superior control, one has to look at the vector control as on a system with variable dynamics (having variable parameters), which complicates the design of the IM control. The dynamics of the load torque compensation in the case with a increased moment of inertia $J = 2 \, J_N$ in time $t = 3$ s has better quality at the vector control. In the proposed control structure, the change of the moment of inertia does not affect the quality of its compensation.

The comparison between the proposed and the vector control structures shows that the vector control structure exhibits a nonlinear behavior, while the proposed structure always shows linear transient responses. In terms of the dynamics, the responses of these structures are very similar to the basic setting of the parameters of the controlled system. However, the robustness of the proposed control structure is significantly better with changes of unknown parameters.

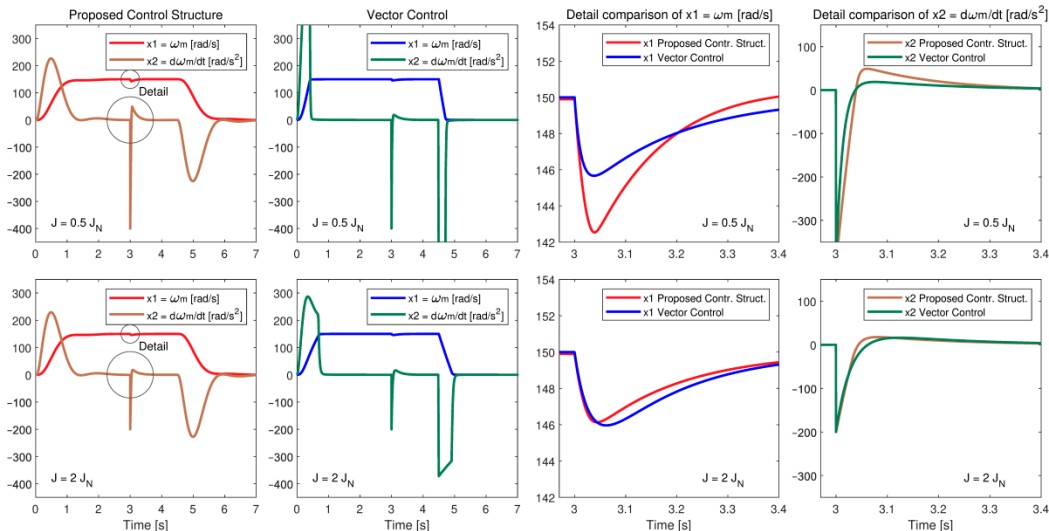

**Figure 13.** Comparison of proposed control structure and vector control for moment of inertia changed to $J = 0.5\,J_N$ and $J = 2\,J_N$.

## 7. Experimental Verification of the IM Drive Speed Control on a Laboratory Model

The linear model reference control structure is verified on the three-phase induction motor TYPE 1AV3104B (400 V, 2.2 kW, 1465 RPM, 14.3 Nm) driven by the VQFREM 400 004–4MA power converter (400 V, 11 A), manufactured by VONSCH Co, Brezno, Slovakia. The structure of the experimental setup is shown in Figure 14.

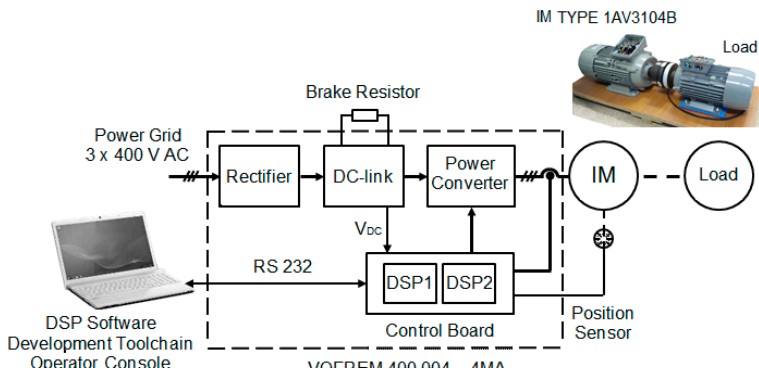

**Figure 14.** The structure of the experimental setup with VQFREM 400 004–4MA power converter.

The control board of the power converter has two processors DSP TMS320F2406. The first DSP processes signals from the current, DC-link, and position sensors. It also performs the Clarke and Park transformations and uses PI type controllers with a sample time of 50 μs to controls the stator currents $i_{1x}$ and $i_{1y}$ in the $\{x, y\}$ reference frame. The second DSP implements the speed control and communicates with a higher-level controller and operator console. The speed control sample time is 1 ms. The fast current control loop is written in assembler language and the rest of the code is written in C. The Texas Instruments software development toolchain is used for programming DSPs. The setup with the input choke, power converter, and brake resistors is shown in Figure 15.

To verify experimentally the proposed control structure in Figure 4, several measurements were performed on a laboratory model with IM at its starting to the rated speed $\omega_m = 150$ rad/s in time $t = 0.5$ s. An influence of the optional positive parameter in the reference model $\alpha$ on the motor dynamics was investigated. For example, Figure 16a shows the control dynamics for the parameter $\alpha = 5$ (Equation (23)) and the values of the optional vector $\mathbf{k}^T$ elements $\mathbf{k}^T = [0.0031\ 0.0019\ 0.00038]$.

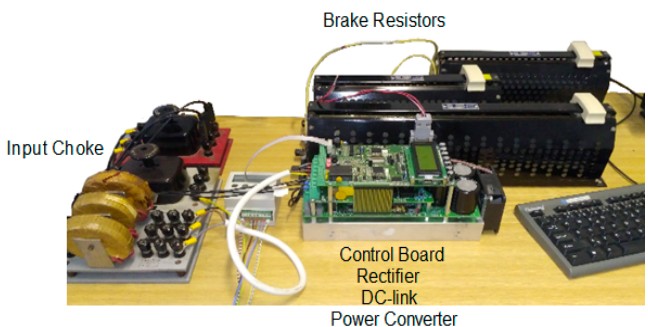

**Figure 15.** The experimental setup with VQFREM 400 004–4MA converter.

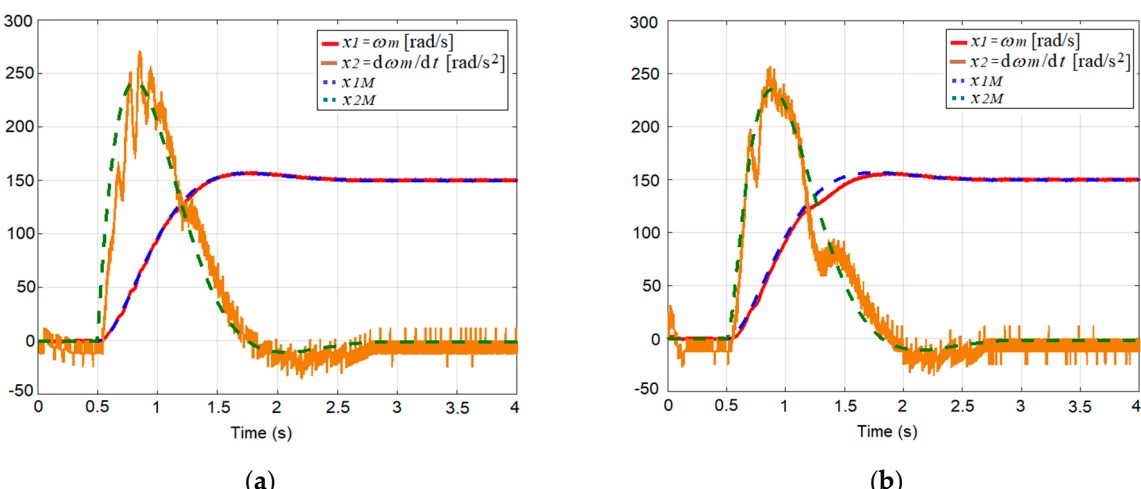

(**a**)          (**b**)

**Figure 16.** Motor starting at: (**a**) $\alpha = 5$ and $\mathbf{k}^T = [0.0031\ 0.0019\ 0.00038]$; (**b**) $\alpha = 5$ and $\mathbf{k}^T = 0.25\ [0.0031\ 0.0019\ 0.00038]$.

In the graphs, the time courses of the model state variables $x_{1M}$, $x_{2M}$, prescribing the required dynamics for the motor angular speed and acceleration are compared with their measured values. The time courses of the measured quantities show a very good agreement with the reference model.

Figure 16b shows time similar courses when decreasing the optional gain of the elements of the vector $\mathbf{k}^T$ to 25% of their original value. It is obvious that the introduced change influences the time course of variables in dynamic states (e.g., here the state variable $x_2$ oscillates less, but it has a larger absolute deviation from the reference model). It is important that again a substantial agreement between the measured quantities and the reference model outputs has been reached.

The time courses in Figure 17a show the dynamics of the IM drive starting at 300% change of the controller optional gain $\mathbf{k}^T$ against its original value. Due to the substantial increase of the gain in the control circuit, the state variable $x_2$ (presenting the acceleration proportional to the motor dynamic torque) is visibly more oscillating, while the output variable $x_1$ (IM speed) practically follows the reference model.

The starting of the motor from the non-energized state and its stopping for the values of the control parameters valid for Figure 16a are shown in Figure 17b (here the simulation was performed for the motor starting in time $t = 0$). One can observe here that the motor follows the reference model both during starting and stopping, i.e., it behaves as a linear system according to the reference model (23). Therefore, the proposed control structure is able to ensure the quality of the angular speed IM control within the whole control range.

The experimental measurements confirm that the controlled drive behaves with high accuracy as its reference model, where for this case an optimal 2nd order linear system was chosen. The reference model dynamics are determined by the optional parameter $\alpha$ and the gain vector $\mathbf{k}^T$.

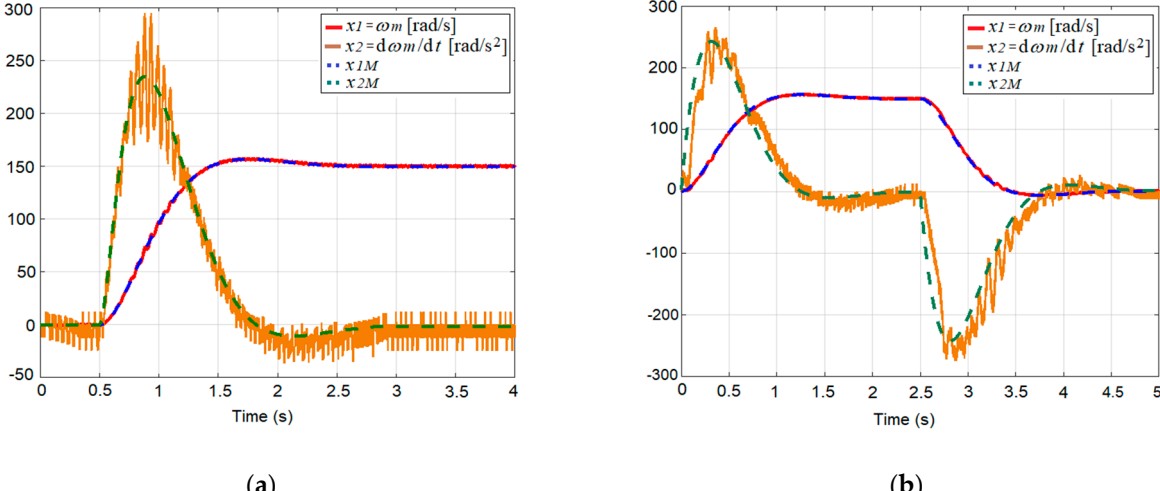

**Figure 17.** Motor starting at: (**a**) $\alpha = 5$ and $\mathbf{k}^T = 3$ [0.0031 0.0019 0.00038]; (**b**) $\alpha = 5$ and $\mathbf{k}^T = [0.0031\ 0.0019\ 0.00038]$.

## 8. Discussion

Based on the simulation and experimental results and on comparison of the designed control structure with the vector control structure, the proposed new control structure with the reference model is characterized by the following properties:

- In general, the order of the reference model must be the same one as the order of the controlled system. In practice, it is often possible to reduce the controlled system order (e.g., in the case of electrical machines, the influence of fast control loops can be neglected, the dynamics of electromagnetic phenomena can also be replaced by the 1st order system, etc.). In the case of the IM, the number of the reference model state variables has been the same as the order of the reduced controlled system. The reference model of the IM presents a nonlinear 2nd order system (Figure 3). This significantly simplifies the control structure while maintaining its required properties which has been proved by the simulation on the 3rd order IM model and by experimentation on a real drive with IM (Sections 5 and 7).

- For the new control structure, no accurate values of the controlled system parameters are required. This fact is described in Section 6, where the influence of an unknown change of the motor important parameters ($R_2$, $J$) on the dynamics is shown in Figures 11–13 and compared performance of the designed control structure and vector control structure. This fact is shown and described in more detail in Section 6, where the influence of the unknown change of important parameters ($R_2$, $J$) on the control dynamics of the designed and vector control structures are compared in Figures 11–13.

- The parameter's setting is performed through two constant parameters (the gains). The first parameter (the parameter $\alpha$ in the description of the reference model (23)) determines the overall control dynamics. It can be determined from the measured standardized responses of the controlled system (e.g., based on unit step characteristics, Figure 2) using the Shanon–Kotelnik theorem. The elements of the vector $\mathbf{k}^T$ that ensure the transient phenomena and stability present further optional parameters. They must comply with condition (16). The setting of the mentioned parameters can be realized without any exact knowledge of the controlled nonlinear system parameters. They must be positive and are tuned by a gradual increase of their values up to physical limitations of the controlled system, i.e., up to the current limitation of the IM.

- Comparing with the speed vector control structure, the proposed control structure is a simpler one that concerns the interconnection and lower order of the control structure). The use of the necessary Park transform in the proposed control structure

is independent from the exact knowledge of the IM parameters which increases the quality of control (as shown in Section 6).

- From the view of the superior control, the novel structure exhibits dynamic properties of an optimal linear dynamic system satisfying the criteria of minimum control deviation and minimum input energy [32]. Thus the drive properties are practically identical to the properties of a high quality vector control of the IM drive.
- In the proposed control structure, the controlled nonlinear system is complemented by linear subsystems only (Figure 8a). For this reason, it can be implemented by cheap conventional hardware.
- The stability of the proposed control structure is ensured by calculating the parameters (the elements of the positive definite matrix **P**) by means of the Lyapunov matrix Equation (11) or in the case of an optimal chosen model according to [32], they can be calculated from (27).
- The proposed control structure is robust against considerable variations changes not only of the important parameters of IM (e.g., of the rotor resistance $R_2$ (Figure 6) or of the moment of inertia—Figure 7a,b), but also to inaccurate known parameters of the nonlinear controlled system. Here, the control system follows with high accuracy the reference model output which is confirmed by the simulations in Figures 11–13 and also by the measurements in Figures 16 and 17.
- The proposed control structure is invariant against the external disturbances (e.g., of the load torque equal to the nominal torque: Figures 5–7) where the dynamics of the disturbance is very fast.
- The proposed control of the IM angular speed solves also the problems occurring at low speed. The controlled drive follows its reference model with high accuracy also in the range of low speeds up to zero speed.

## 9. Conclusions

The designed new stable control method can be applied generally for any continuous nonlinear system. In this article, a controlled AC drive with an induction motor was chosen for verification of its properties, where the IM represents a highly oscillating nonlinear system and its parameters (e.g., the rotor resistance and moment of inertia) can vary during the motor operation.

Both simulation results and experimental measurements performed for basic operating states of the IM drive have confirmed advantages of the proposed controller concerning simplicity at the design and implementation and the excellent performance of the controlled system. Application of the new control method considerably contributes to improving dynamic properties and simultaneously ensures the drive stability, invariance against disturbances and robustness against the motor parameters variations. Compared to the control structures of the vector control of IM, the new control structure is significantly simpler; it exhibits features of an optimal linear system and simultaneously it achieves almost identical dynamic control performance. When comparing with the classical vector control structure of the IM, the designed control structure is much more robust against changes of the important motor parameters.

The main advantage of the novel control structure consists of the fact that it does not require any knowledge of controlled system parameters. On the other side, the precise control at the majority of the used control methods strongly depends on precise knowledge of system parameters. The control quality is also guaranteed by the independence of the Park transformation calculation from the exact knowledge of the IM parameters. The control structure ensures high quality of the IM speed control within the whole control range, up to zero speeds.

The proposed control structure is suitable for the control of continuous nonlinear systems with unknown and time-varying parameters. The controller is suitable especially for any drive system including control of robotic systems control having a precise hierarchical control structure. Therefore, its broad utilization in industrial applications can be assumed.

**Author Contributions:** Conceptualization, P.F. and D.P.; methodology, P.F. and D.P.; writing—original draft preparation, P.F.; writing—review and editing, D.P.; formal analysis, P.B.; software, P.B. and M.F.; validation, P.B. and M.F. All authors have read and agreed to the published version of the manuscript.

**Funding:** This work was supported by the Slovak Research and Development Agency under the contract No. APVV-19-0210.

**Institutional Review Board Statement:** Not applicable.

**Informed Consent Statement:** Not applicable.

**Data Availability Statement:** Not applicable.

**Conflicts of Interest:** The authors declare no conflict of interest.

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
