# Peer review of "New Stable Non-Vector Control Structure for Induction Motor Drive"

_applsci, doi:10.3390/app11146518_

Round 1

Reviewer 1 Report

The parameter variation in IMs is not such an important problem, as the slip itself can compensate for the variations.

A comparison between the proposed method and vector control is done, yet, no performance comparison or cuantitative results are shown, nor in simulation neither experimentally. Only structure is compared.

A comparison againts closed loop scalar control should also be considered.

The ease of tunning the control parameters compared to more classical approaches is not evident.

In overall, a proper comparison is needed to clearly show how the proposed method improves the traditionally used ones.

Reviewer 2 Report

This paper presents the design, simulation and experimental validation of a nonlinear control systems based on a linear reference model. The main idea and the paper novelty consists in the development of the proposed control structure using a newly introduced state variable. The presented solution is validated considering the angular speed control of an induction motor drive as a nonlinear controlled system. It should be noted that, in addition to the validation by simulation, the solution is also experimentally tested. Also, the system stability is analyzed on basis of the Lyapunov's second method.

The conclusions are relatively clear, highlighting the contributions of the proposed control strategy. One observation in conclusion: if you say -compared with the classical vector control strategy, control performance is almost identical-, this must be proved by results. The paper is well written and the issues are well described.

However, I have a few questions and recommendations to improve the manuscript:

- Chapter 6 (Comparison of the Proposed Control Structure with the Vector Control Structure) does not present comparative results regarding the proposed control strategy, respectively vector control strategy. I think that such a performance comparison of these two control strategies would be useful.

- Regarding the results from Fig. 2, what is the time moment when the step change on the motor inputs occurs? Is this time moment t = 0 s? These time moments (when a disturbance acts, a parameter changes, etc.) should also be specified for the other presented results of the performed validation (chapter 7).

- How they were chosen and set the α (alfa) parameter (chapter 5 and 7) and the controller gain kT (chapter 5)?

- The controlled process is modeled by a 3rd order nonlinear system (which is not such a higher order). To design the proposed control strategy (chapter 4), a simplified 2nd order nonlinear system was used. What simplifying assumptions were taken into account to reduce the order? Why was not used the 3rd order complete model?

Round 2

Reviewer 1 Report

All the previously identified issues where correctly addressed, showing the advantages of the proposed method a lot more clearly.